# Microstructures and Phases in Electron Beam Additively Manufactured Ti-Al-Mo-Zr-V/CuAl9Mn2 Alloy

**DOI:** 10.3390/ma16124279

**Published:** 2023-06-09

**Authors:** Anna Zykova, Aleksandra Nikolaeva, Aleksandr Panfilov, Andrey Vorontsov, Alisa Nikonenko, Artem Dobrovolsky, Andrey Chumaevskii, Denis Gurianov, Andrey Filippov, Natalya Semenchuk, Nikolai Savchenko, Evgeny Kolubaev, Sergei Tarasov

**Affiliations:** Institute of Strength Physics and Materials Science, Siberian Branch of Russian Academy of Sciences, Tomsk 634055, Russia; zykovaap@ispms.ru (A.Z.); nikolaeva@ispms.ru (A.N.); alexpl@ispms.ru (A.P.); vav@ispms.ru (A.V.); aliska-nik@mail.ru (A.N.); artdobrov@mail.ru (A.D.); desa-93@mail.ru (D.G.); avf@ispms.ru (A.F.); natali.t.v@ispms.tsc.ru (N.S.); savnick@ispms.ru (N.S.); eak@ispms.ru (E.K.); tsy@ispms.ru (S.T.)

**Keywords:** electron beam additive manufacturing, in situ composite, composite aluminum bronze/Ti, phase transformation, Cu-Ti-Al, mechanical properties, wear resistance

## Abstract

Electron beam additive manufacturing from dissimilar metal wires was used to intermix 5, 10 and 15 vol.% of Ti-Al-Mo-Z-V titanium alloy with CuAl9Mn2 bronze on a stainless steel substrate. The resulting alloys were subjected to investigations into their microstructural, phase and mechanical characteristics. It was shown that different microstructures were formed in an alloy containing 5 vol.% titanium alloy, as well as others containing 10 and 15 vol.%. The first was characterized by structural components such as solid solution, eutectic intermetallic compound TiCu_2_Al and coarse grains of γ_1_-Al_4_Cu_9_. It had enhanced strength and demonstrated steady oxidation wear in sliding tests. The other two alloys also contained large flower-like Ti(Cu,Al)_2_ dendrites that appeared due to the thermal decomposition of γ_1_-Al_4_Cu_9_. This structural transformation resulted in catastrophic embrittlement of the composite and changing of wear mechanism from oxidative to abrasive.

## 1. Introduction

Additive manufacturing is a rapidly growing area in almost all industries, but traditionally, the most demanded applications of these methods relate to the fabrication of metallic components [1,2,3,4]. While many alloys and composites are made using traditional casting or powder manufacturing methods, additive manufacturing offers new solutions for in situ control and modification of the alloy microstructure [5,6,7,8,9,10,11]. Electron beam wire additive manufacturing is a highly productive process that allows simultaneous additive deposition of dissimilar metals and in situ forming a composite [1,12,13]. Structures and phases that result from using additive methods nucleate and grow under conditions that are different from those in casting. Layer-by-layer deposition results in cyclically reheating the part of the as-deposited underlying metal and forming heat-affected layers. Therefore, additively built metal is subjected to cyclic heat treatment, which provokes microstructural changes. As a result, complex structural phase states may be formed that determine mechanical characteristics of the samples built from several components, including immiscible ones such as Fe and Cu or stainless steel–copper [14].

Copper and titanium have different mechanical properties, which make it possible to create alloys with different characteristics of strength, hardness, and deformation properties to be used in various industries such as aviation, space, electronics, and others. The Cu-Ti system is the object of active research in order to create new materials with unique properties.

The relatively new potential of Ti-Cu alloys is related to the microbiological activity of copper–titanium alloys [15,16]. Combined biomedical/strengthening effect of copper addition to titanium alloy may be achieved by alloying titanium alloys with copper and precipitation of Ti_2_Cu intermetallic compound particles [17,18].

On the other hand, the dissolution of the Ti_2_Cu in solution treatment at 800 °C allowed a low-elastic-modulus biomedical Ti-6Al-4V-5.6Cu alloy to be obtained [19] with improved corrosion resistance. One of the findings here that allowed improved yield strength/Young modulus ratio of the alloy was the effect of martensite reorientation-induced plasticity caused by α″ transformation into α′ with its further reorientation.

The use of aluminum bronze instead of pure copper for alloying titanium alloys would allow more effective intermixing of melts and enable a diversity of phases to be obtained in addition to Ti_2_Cu due to the better wettability of Ti by the aluminum bronze. Aluminum bronzes offer high strength and ductility, excellent thermal conductivity, and wear and corrosion resistances [20]; therefore, they are widely used in the aerospace, electronics and automotive industries. Increasingly, aluminum bronze is being researched and considered for use as a wear-resistant material due to its outstanding friction and wear reduction properties. In some cases, aluminum bronze is applied to various alloys as a coating [21,22], while in other applications, it is used as a component of composites [23,24,25,26,27,28,29]. In particular, more and more attention is being drawn to composite materials with intermetallic reinforcement based on aluminum bronze to be a kind of wear-resistant material.

Li Wen-Sheng et al. [23] developed a new composition of aluminum bronze Cu-14%AI-X containing microadditives of elements with the aim of improving the mechanical characteristics. The additives formed fine phases homogenously dispersed in the bronze matrix that allowed improving the matrix bronze characteristics greatly.

The effect of Ti on the microstructure and wear behavior of a selective laser-melted Al-Cu-Mg-Si alloy was studied [30]. It was found that with the addition of Ti, the wear rate of SLM processed Al–Cu–Mg–Si decreased by 49.6% for the applied load of 10 N. The refined heterogeneous microstructure helps to achieve strength-ductility synergy of the Al–Cu–Mg–Si–Ti alloy, which provides superior resistance to indentation behavior and initiation of cracks. The addition of Ti allows the wettability of a matrix alloy to be improved with respect to nitrides, carbides and oxides.

The tribological properties of the ternary intermetallic compound TiCu_2_Al were studied [31]. It was shown that the coefficient of friction of the coating prepared from the TiCu_2_Al intermetallic compound reduced with an increase in the normal load and sliding speed. The wear rate of the TiCu_2_Al intermetallic coating rapidly decreased with increasing sliding speed, while the wear rate first increased and then decreased under normal load from 5 to 15 N.

Alloys were prepared using Ti-6Al-4V and C11000 copper wires via wire arc additive manufacturing (WAAM) [32]. It was found that both formation of the CuTi intermetallic compound and the transformation of the β phase depend on the heat input. The Ti-Cu alloy may be easy to process and possess good corrosion resistance, which is important for fabricating components working in aggressive media [32].

To date, a limited number of studies have been carried out on the effect of various concentrations of Ti on the wear resistance of aluminum bronze, there are no publications on the preparation of such composites by additive methods, and therefore, further research is needed. Thus, the purpose of this work was to study the effect of titanium additives on the evolution of the structural phase composition, mechanical and tribological properties of aluminum bronze.

## 2. Materials and Methods

Alloy CuAl9Mn2/VT-20 walls were prepared using an electron beam additive manufacturing (EBAM) machine (Institute of Strength Physics and Materials Science of Siberian Branch of Russian Academy of Sciences, Tomsk, Russia) attached to a pair of wire feeding devices, as shown in Figure 1a. The element compositions of aluminum bronze CuA19Mn2 and titanium alloy VT-20 (Ti-Al-V-Mo-Zr) Ø1.6 mm wires are shown in Table 1. These wires were simultaneously fed into a molten pool created by an electron beam firstly in the substrate or later in the previously deposited CuAl9Mn2/VT-20 alloy. The substrate was machined from an ASTM 321 stainless steel. The first layers were deposited using electron beam current 65 mA to provide better fusion and a defectless transition zone between the composite and the cold substrate. Later, when depositing the basic part of the wall, the beam current was gradually reduced to 42 mA to avoid overheating. Accelerating voltage was maintained constant at 30 kV throughout the full manufacturing. The substrate was moved horizontally at 400 mm/min during deposition, while the percentage of either bronze or titanium alloy in the alloy was maintained by adjusting the wire feed rates. Three types of alloy were produced according to the above-described technique: bronze/5%Ti, bronze/10%Ti and bronze/15%Ti, with the component concentration CuAl9Mn2/VT-20 ratios as follows: 95:5, 90:10, 85:15. Full elemental compositions of the resulting alloys were obtained using an X-ray fluorescence (XRF) spectrometer Niton XL3t 980 GOLDD and are presented in Table 1.

Samples were cut from the wall to perform microstructural examination and to determine their mechanical characteristics, as shown in Figure 1b. Metallographic views were prepared using a standard procedure, including grinding, polishing and etching in a solution of 30 mL HCl + 5 g FeCl_3_-6H_2_O + 60 mL H_2_O. An optical microscope Altami (Altami Ltd., Saint Petersburg, Russia) was used to obtain the CuAl9Mn2/VT-20 macrographs. Microstructural characterization was carried out using a high-resolution field emission scanning electron microscope (SEM) QUANTA 200 (EDAX, Netherlands) with an EDS add-on attached. A TEM instrument JEOL (Tokyo, Japan) and X-ray dractometer XRD-7000S (Co_Kα_) (Yekaterinburg, Russia) were used to study the fine structures and phases of the additively manufactured alloys.

Mechanical characteristics of the samples were studied using a Vickers microhardness tester TBM 5215A Tochline (GK “Tochpribor”, Ivanovo, Russia) and a tensile machine UTS-110M. The tensile test standard dog-bone specimens were machined from the walls so that their tensile axes were oriented along either the X or Z axis (Figure 1b), thus representing the layer deposition and wall building directions, respectively. Three specimens were tested for each experimental datapoint.

Tribological testing was carried out using a “ball-on-disk” scheme, where a Si_3_N_4_ Ø6 mm ball was sliding on a disk machined from the alloy wall without adding any lubricant (Figure 1b). Sliding speed and path length were 100 mm/s and 400 m, respectively. Testing was carried out with the normal force values 9, 14, and 25 N. Wear was evaluated by measuring the cross-section area of the wear track using a laser scanning microscope Olympus OLS LEXT 4100.

To characterize the dynamics of sliding acoustic emission (AE) and vibration acceleration (VA) signals were registered using an AE setup EYa-1 (Togliatti State University) and commercial accelerometer IMI attached with data registration device USB NI-9234 (National Instruments) with sampling frequencies 6.25 MHz and 25.6 kHz, respectively.

## 3. Results

### 3.1. Structures and Phases in the Bronze/5%Ti Alloy

The microstructure views and corresponding XRD patterns in Figure 2 represent the microstructural and phase composition evolution of the bronze/5%Ti alloy as observed along the vertical OZ axis from the substrate to the top (Figure 1b). According to the XRD patterns, the as-deposited CuAl9Mn2/VT-20 part is composed of α-Cu solid solution and intermetallic compounds such as Al_4_Cu_9_ and TiCu_2_Al (Figure 2c,f,i). Metallographic images show the α-Cu in the form of dendritic structures without any preferential orientation of their primary axes (Figure 2a,d,g), while Al_4_Cu_9_ grains differ from them by whitish contrast (Figure 2b). It should be noted that the number of these whitish grains decreases in the middle and bottom parts, compared to in the top part. The interdendritic spaces contain eutectic structures (Figure 2b,e,h) whose content increases when viewed from the top to the bottom part of the wall. The α-Cu dendrites contain some coarse particles whose size also increases when approaching the bottom part of the wall.

A transition zone was formed between the as-deposited CuAl9Mn2/VT-20 and stainless steel substrate where all components intermixed (Figure 2j,k,l) and formed phases such as α-Cu, α-Fe, γ-Fe, Fe_2_Ti and Al_2_Cu (Figure 2l). According to XFR (Table 1), the as-deposited bronze/5%Ti metal contained 0.3 ± 0.01 wt. % Fe and 0.3 ± 0.01 wt. % Ni. Despite both source wires containing some minor concentrations of these elements, the high percentages of both metals make it possible to suggest that they came from the substrate and were evenly distributed in the wall bulk by means of EBAM layer-by-layer deposition, partial melting, and convection in the molten pool.

The α-Cu dendrite mean sizes in the top, middle and bottom parts of the wall were 15.9 μm, 15.3 μm, and 10.1 μm, respectively (Figure 2b,e,h).

TEM images of thin foils prepared from the middle part of the bronze/5%Ti walls observing α-Cu grains with numerous dislocations, stacking faults (Figure 3a–c) and~200 nm precipitations identified as Ti_x_Cu_y_Al_z_ according to EDS elemental composition (Figure 3d, Table 2, spectra 1–4). Some of them may contain up to 6 at.% Fe (Figure 3d, Table 2, spectrum 3). Spectrum 5 (Figure 3d, Table 2) corresponds to that of coarse α-Cu grains. Other structural components found in the as-deposited bronze/5%Ti metal are TiCu_2_Al (Figure 3e,f, Table 2, spectrum 6 and Figure 4a,b, Table 2, spectrum 7) and Al_4_Cu_9_ (Figure 4a,c, Table 2, spectra 8,9). These Al_4_Cu_9_ coarse grains contain 33–37 at.% Al, and ~63–67 at.% Cu annealing twin boundaries (Figure 4a,d–f). Intermetallic FCC TiCu_2_Al grains contain ~20 at.% Ti, 35–39 at.% Al and 39–41 at.% Cu (Table 2, spectra 6,7).

The volume percentages of (α + Al_4_Cu_9_ + TiCu_2_Al) eutectics in the top, middle and bottom parts of the wall were 15, 20 and 25%, respectively. The balance is accounted for by the aluminum bronze matrix. The mean size of the TiCu_2_Al grains is constant at ~3 μm, irrespective of the location.

### 3.2. Structures and Phases in the Bronze/10%Ti Alloy

More diversified and complicated structures were formed in a bronze/10%Ti alloy (Figure 5a,b,d,e,g,h) with the participation of phases the same as those detected in the bronze/5%Ti one, i.e., α-Cu, Al_4_Cu_9_ and TiCu_2_Al (Figure 5c,f,j). First of all, this sample shows a kind of macrostructural inhomogeneity, as observed along the wall’s height. The bottom part visually contains more eutectics with fine intermetallic particles (Figure 5j) in comparison to the middle and top parts (Figure 5a,d,g). Other structural components that are present in both the bottom and middle parts are coarse equiaxed and flower-shaped dendritic particles (Figure 5d,e,g,h). It may be the case, however, that the first ones appear as segments of the latter ones, sectioned by the metallographic view plane.

The middle and, especially, the top parts are characterized by more areas occupied by the α-Cu grains and less areas occupied by eutectics and coarse particles (Figure 5a,b,d,e). The flower-like particles are also observed in the transition zone of the bronze/10%Ti alloy (Figure 5e,h), together with high-aspect-ratio grains (Figure 5k,l); however, their inner parts look different from those found inside the bottom and middle parts. The EDS profile of one of them, presented in Figure 6a,b, shows its composition to be Fe-Ti (Figure 6c), provided that the Fe curve has a minimum between two maxima in the center of the particle coinciding with the small maximum of the Cu curve. It can be suggested that this particle has a Ti_x_Cu_1−x_ core surrounded by a Fe_x_Ti_1−x_ shell. All other elements, such as Cr, Ni, Mn, V, and Al, can also be found in this particle, judging by the spectra in Figure 6c. The XRD pattern in Figure 5m indicates the presence of Fe_2_Ti in the transition zone in addition to α-Cu, α-Fe, γ-Fe and Al_2_Cu.

The flower-like dendritic particles found in the middle part of the bronze/10%Ti show the presence of mainly Ti and Cu and tiny peaks of elements like Al, Fe, Mn, and V (Figure 6c–f). It seems that these dendrites belong mainly to the Ti-Cu system, and may be an onset stage of Ti(Cu,Al)_2_ formation, whose XRD peaks may be identified from corresponding XRD patterns (see insets to Figure 5c,f).

The α-Cu grains in the bronze/10%Ti contain small Al_4_Cu_9_ grains (Figure 7a–e). Their volume content is low enough not to be detected in either the middle or the bottom parts by means of XRD (Figure 5c,f,j). In addition to coarse ones, there are small 1.5 μm Ti-Cu-Al particles (Figure 7a,c,e) containing 33.9 at.% Al, 20.7 at.% Ti and 42.2 at.% Cu (Figure 8a,b), which can be identified as Ti(Cu,Al)_2_ according to the SAED pattern in Figure 8c. This Ti(Cu,Al)_2_ phase is characterized by a hexagonal crystalline lattice with parameters a= 0.5013 nm and c = 0.8122 nm. The EDS elemental maps in Figure 8b,d–i testify that both Fe and Ni are contained in these grains. In addition, the bronze/10%Ti alloy contains grey particles localized in the vicinity of TiCu_2_Al and Ti(Cu,Al)_2_ particles (Figure 5b,e,h and Figure 8b). These initially hexagonal Ti-Si-Mo-V particles (Figure 9) appear to be partially dissolved with the inside boundary between at least two grains separating the inside half-circle Ti-lean area from the Ti-rich one (Figure 9f). Presumably, these Ti-lean and Ti-rich inner grains could be Al_4_Cu_9_ and TiCu_2_Al grains grown from the dissolution of the Ti-Si-Mo-V-Ni hexagonal grains under cyclic reheating.

### 3.3. Structures and Phases in the Bronze/15%Ti Alloy

A more homogeneous microstructure was formed in the bronze/15%Ti alloy parts, with mainly eutectic structures and intermetallic flower-like dendrites (Figure 10a,b,d,e,g,h). The XRD patterns enable the identification of the presence of α-Cu, Al_4_Cu_9_ and TiCu_2_Al phases (Figure 10c,f,j) in all three as-deposited parts. Additionally, very small Ti(Cu,Al)_2_ peaks can be distinguished in these XRD patterns (see insets to Figure 10c,f,i).

The transition zone shows intermetallic particles with both platelet and flower morphologies (Figure 10j,k), and phases such as α-Cu, α-Fe, γ-Fe, and Fe_2_Ti (Figure 10m).

TEM examination of thin foils showed the presence of Ti(Cu,Al)_2_ (Figure 11a) hexagonal particles containing 39 at.% Si, 45 at.% Ti, 3.2 at.% V, 4.3 at.% Cu and 8.7 at.% Mo (Figure 11b), and α-Cu grains with annealing twin boundaries (Figure 11c,d,f). Small intermetallic particles inside eutectic areas were identified as TiCu_2_Al (Figure 11c,e). The flower-like particles found in this part were identified as Ti(Cu,Al)_2_ particles, with admixed elements such as V, Fe Ni and Mn (Figure 11a).

### 3.4. Mechanical Characteristics of CuAl9Mn2/VT-20 Alloy

The microhardness profiles of CuAl9Mn2/VT-20 alloy were obtained by creating indentations along a line parallel to the wall height (Figure 12). All these microhardness profiles were characterized by the extent of scatter in the microhardness numbers resulting from the structural inhomogeneity of the alloy samples. This scatter was less wide for the bronze/5%Ti sample, which had a mean matrix microhardness at a level of 1.70 ± 0.01 GPa, with higher peaks resulting from the indentation of the TiCu_2_Al areas. The scatter was wider for the bronze/10%Ti sample, where all three parts showed different levels of structural inhomogeneity. The middle part of this sample was structurally represented by areas with intermetallic dendrites Ti(Cu,Al)_2_, which enhanced the hardness to 2.3 GPa. An even higher microhardness was achieved when increasing the titanium alloy concentration to 15 vol.%.

The dependencies of ultimate tensile strength (UTS) and relative elongation ε on the titanium alloy percentage show that neither of these characteristics are high, and are impaired by higher titanium alloy concentrations (Figure 13). The maximum UTS and ε values ~500 ± 32 MPa and 2–6%, respectively, were obtained for the bronze/5%Ti alloy structurally composed of α-Cu dendrites and small eutectic particles. All the alloys demonstrated anisotropy of their mechanical characteristics, depending on the tensile axis orientation with respect to building direction. The hor1 samples cut from the bottom part demonstrated almost linearly decreasing UTS vs. VT-20 percentage dependence. The hor2 samples cut from the top part showed increased UTS at 10% VT-20, followed by at sharp fall at 15% VT-20.

A large difference was observed for UTS in samples vert1 and vert2 oriented with their tensile axes along the deposition layer direction but cut from the bottom (vert1) and top (vert2) parts of the wall at the VT-20 concentration of 5%. This difference can be explained by microstructural differences between the top and bottom parts (Figure 2). Higher amounts of eutectics in the bottom part of the wall serve to impair vertical strength. At the same time, the horizontal strength of the bottom part metal (hor1) is higher than that of the top part (hor2). Adding 10 and, especially, 15 vol.% titanium alloy resulted in catastrophic embrittlement of the samples.

### 3.5. Tribological Testing

Mean values of the coefficient of friction (CoF) were determined at the steady sliding stages after a running-in period. The dependence of CoF on the normal load for the bronze/5%Ti sample demonstrates its almost constant value in the range 0.19–0.22 (Figure 14a). Two other dependencies show CoF values of 0.24–0.34 and 0.31–0.37 for bronze/10%Ti and bronze/15%Ti samples, respectively. Some reduction in CoFs can be observed at normal load values of 14 and 19 N. It should be noted that wear was quantified by measuring the wear scar cross-section areas (Figure 14b).

## 4. Discussion

### 4.1. Structural Phase Formation

The formation of the complicated inhomogeneous structural phase composition of CuAl9Mn2/VT-20 alloys is a result of several phenomena and technological features of electron beam additive manufacturing. Firstly, due to the use of a steel substrate, the dilution and partial dissolution of its alloying elements with those of the molten titanium alloy and aluminum bronze occurs. Dissolution primarily concerns iron, nickel and chromium, which can form several intermetallic compounds. Secondly, all deposited layers are sequentially heated and cooled with decreasing amplitude for each new layer deposited on their top. With each additional layer, cyclic reheating and cooling occurs, which leads to the modification of the structural phase composition. Consequently, the local properties of the material of the finally constructed product will depend on the process parameters and the formed structure.

It has been previously established that the EBAM aluminum bronze CuAl9Mn2 is composed of α-Cu dendrites and inter-dendrite β′-(Cu_3_Al) [33]. The addition of 5 vol.% of titanium alloy into aluminum bronze resulted in the formation of structures composed of α-Cu dendrites, γ_1_-Al_4_Cu_9_ grains and inter-dendrite TiCu_2_Al precipitates instead of the β′ phase.

In the literature, there are two suggestions regarding the formation of TiCu_2_Al. The first one relies on a decomposition mechanism that is similar to that of β phase decomposition according to the Al-Cu diagram. In other words, it is suggested that Ti has a stabilizing effect on the β phase until the temperature interval 560–500 °C is reached, and then the latter is decomposed into α-Cu, γ_1_-Al_4_Cu_9_ and TiCu_2_Al [34]. Another suggestion is that TiCu_2_Al precipates directly in a copper-rich molten pool at temperatures above 850 °C [34,35,36].

The EBAM samples clearly showed the presence of primary coarse γ_1_-Al_4_Cu_9_ grains. Many literature sources indicate the preferential formation of such intermetallic compounds in Al-Cu systems by mechanical alloying, diffusion, friction stir welding, etc. However, the reasons remain unknown. According to [37], the driving force for the formation of this phase at 773 K is about −47 kJ/mol), i.e., the most negative among the Cu-Al IMCs. One of the possible routes for the formation of this phase can be seen in a diagram in [34], where β-Cu_3_Al precipitates from the molten Ti-Cu-Al metal together with TiCu_2_Al and then decomposes into γ_1_-Al_4_Cu_9_. However, it can be seen from (Figure 5c,f,h and Figure 10b,e,h) that γ_1_-Al_4_Cu_9_ nucleated and grew as dendrites, which is possible only if there was a liquid phase. This is hardly possible, according to the diagram. In addition, it has been shown [37] that the EBAM aluminum bronze is stable against decomposition in cooling β′ phase instead of the β phase.

A possible solution to this problem might be if β-Cu_3_Al → γ_1_-Al_4_Cu_9_ decomposition is accompanied by exothermic heat release and local melting with ensuing dendrite solidification. A similar liquation effect with the formation of flower-like dendrites was observed in friction stir welding on a solid Cu-Al system [38].

Those flower-like dendrites (Figure 5c,f,h and Figure 10b,e,h) look inhomogeneous with whitish inner and dark peripheral regions. Let us assume here that those whitish regions are those of γ_1_-Al_4_Cu_9_, while the dark ones are the products of their thermal decomposition by cyclic reheating. In particular, γ_1_-Al_4_Cu_9_ may decompose into either θ-Al_2_Cu or even α-Cu and Al in heating, so that free Al atoms will diffuse into neighboring areas and form Ti(Cu,Al)_2_. The low thermal stability of mechanically alloyed Al-Cu samples with γ_1_-Al_4_Cu_9_ was shown when heating above 180 °C resulted in the disappearance of its XRD peaks [39].

It is worth noting that the α-Cu bronze matrix contains ~9–11 wt.% of Al, i.e., the same percentage as the source wire, while titanium alloy contains up to 3.7 wt.% of Al. Therefore, the thermal decomposition of γ_1_-Al_4_Cu_9_ may be an effective source of the Al atoms to precipitate Ti(Cu,Al)_2_ according to the following reaction:$$\alpha -\mathrm{Cu}+2\gamma_{1}-{\mathrm{Al}}_{4}{\mathrm{Cu}}_{9}+8{\mathrm{TiCu}}_{2}\mathrm{Al}\;\overset{T,℃}{\to }\;10\alpha -\mathrm{Cu}\;+\;\gamma_{1}-{\mathrm{Al}}_{4}{\mathrm{Cu}}_{9}\;+\;4{\mathrm{TiCu}}_{2}\mathrm{Al}\;+\;4\mathrm{Ti}{\left(\mathrm{Cu},\mathrm{Al}\right)}_{2}$$

Another fact is that the amount of γ_1_-Al_4_Cu_9_ decreased when adding more titanium alloy to the bronze. Thermal decomposition of γ_1_-Al_4_Cu_9_ was also observed in heat treatment of the aluminum bronzes [40,41].

Transition zones contain stainless steel components, and therefore more phases were detected there, including newly formed ones such as α-Fe and Fe_2_Ti. Different types of transition zone microstructures were formed in the alloys containing 5 vol.% and 10–15 vol.% of titanium alloy. The first type of transition zone microstructure, depicted in Figure 2k, was composed of stainless steel particles in the bronze matrix. It is known [42] that stainless steel γ-Fe grains may lose nickel in favor of bronze, and thus partially or fully transform into α-Fe and then into Fe_2_Ti.

Alloys containing 10 vol.% of titanium alloy demonstrated their transition zones to be composed mainly of bright bands and flower-like dendrites, while the 15 vol.% alloy demonstrated only flower-like dendrites in the bronze matrix. The EDS showed them to be composed of Fe and Ti.

### 4.2. Mechanical Characteristics

It follows from the results discussed above that the addition of titanium alloy led to the formation of at least three types of IMCs in the bronze matrix whose contents increased with the amount of titanium alloy added. This structural evolution affected the microhardness, tensile strength and wear resistance of the alloy samples. The microhardness increased with the content of IMCs (Figure 12), while tensile strength and relative elongation decreased because of catastrophic embrittlement (Figure 13).

The unlubricated sliding behavior of these samples showed that CoFs values also increased with the amount of IMCs in the matrix for all normal loads applied (Figure 14a). Rather, the CoF values of the bronze/5%Ti alloy may be related to the anti-friction properties of the bronze matrix reinforced with small IMCs, which provided a steady wear regime characterized by constant wear magnitude at all normal loads. The bronze/10%Ti and bronze/15%Ti alloys contained many more IMCs that might fracture, and thus formed hard wear particles which, in turn, would indent and scratch worn surfaces without allowing them to form an anti-friction oxide film. This suggestion is supported by an analysis of the wear vs. normal load plots (Figure 14b). The bronze/5%Ti alloy showed steady wear dependence on the normal load, while the wear of the two other alloys increased with the normal load.

## 5. Conclusions

Ti-Al-Mo-Z-V/CuAl9Mn2 alloys were prepared using electron beam wire additive manufacturing containing 5, 10 and 15 vol.% titanium alloy.

The bronze/5% alloy was characterized by solid-solution grains with interdendrite small eutectic TiCu_2_Al and coarse γ_1_-Al_4_Cu_9_ particles. New phase Ti(Cu,Al)_2_ dendrites were detected in both bronze-10 vol.% Ti and bronze-15 vol.% Ti alloys, which appeared following the thermal decomposition of γ_1_-Al_4_Cu_9_ particles into α-Cu and Al under cyclic reheating from the layer-by-layer deposition. This resulted in embrittlement of the alloys and the loss of tensile strength and ductility, and the transition of the wear mechanism from oxidation to abrasion.

The addition of titanium alloy led to the formation of at least three types of IMCs in the bronze matrix, the contents of which increased with the amount of titanium alloy added. Such a structural evolution had an effect on the microhardness, tensile strength and wear resistance of the alloy samples.

## Figures and Tables

**Figure 1 materials-16-04279-f001:**
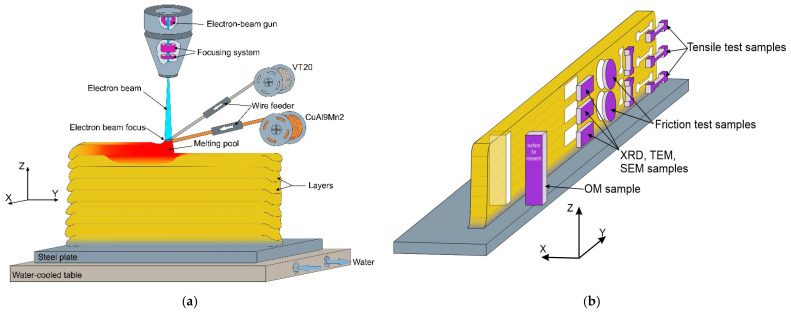
The EBAM diagram with simultaneous feeding two wires used for preparation of CuAl9Mn2/VT-20 alloy walls (**a**) and scheme of sampling for characterization (**b**).

**Figure 2 materials-16-04279-f002:**
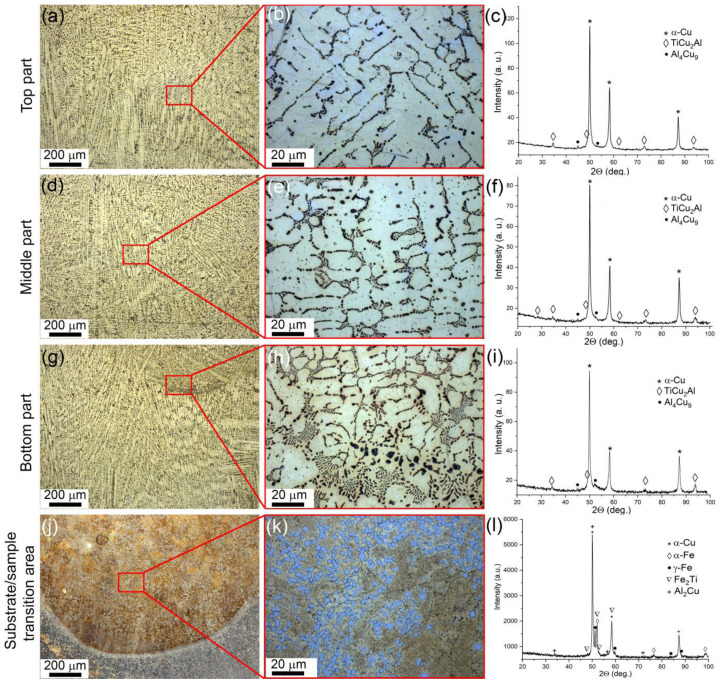
Optical images of macro- and microstructures found in the top (**a**,**b**), middle (**d**,**e**), and bottom (**g**,**h**) parts of the bronze/5%Ti wall and the transition zone (**j**,**k**) with corresponding XRD patterns (**c**,**f**,**i**,**l**).

**Figure 3 materials-16-04279-f003:**
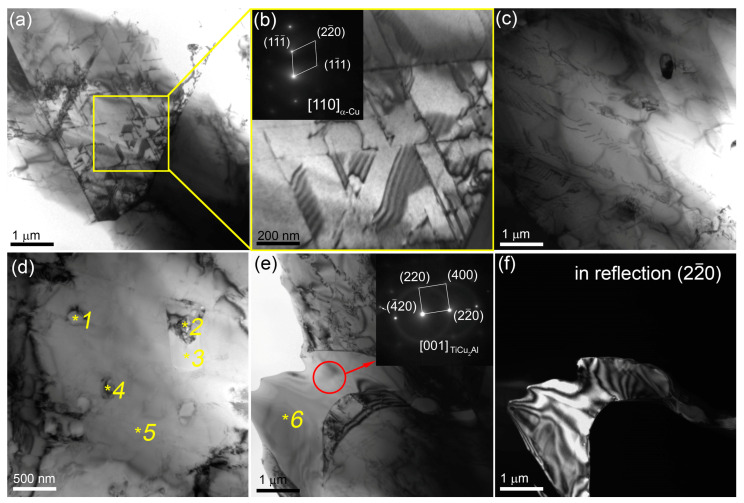
TEM bright-field (**a**–**e**) and dark-field ($2\bar{2}0$)_TiCu2Al_ (**f**) images of fine structures in the middle part of the bronze/5%Ti wall. The numbers *1–*6 denote points at which EDS spectra were obtained (Table 2).

**Figure 4 materials-16-04279-f004:**
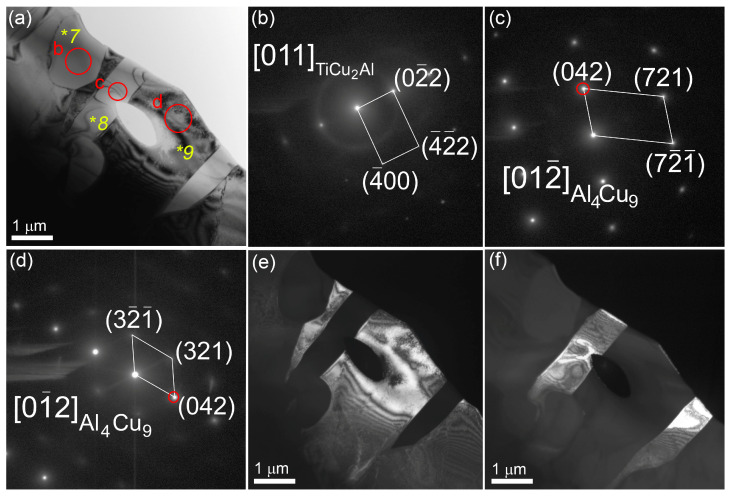
TEM bright-field (**a**), SAED TiCu_2_Al (**b**) and Al_4_Cu_9_ patterns (**c**,**d**) with dark-field images (**e**,**f**) of grains in the middle part of the bronze/5%Ti wall obtained using reflections (042)_Al4Cu9_ indicated by red circles in SAED patterns (**c**,**d**). Numbers *7–*9 stand for points where the EDS spectra have been obtained (see spectra 7–9 in Table 2). Red circles and letters *b–*d show the positions of selector diaphragm used for obtaining SAED patterns in figures (**b**–**d**).

**Figure 5 materials-16-04279-f005:**
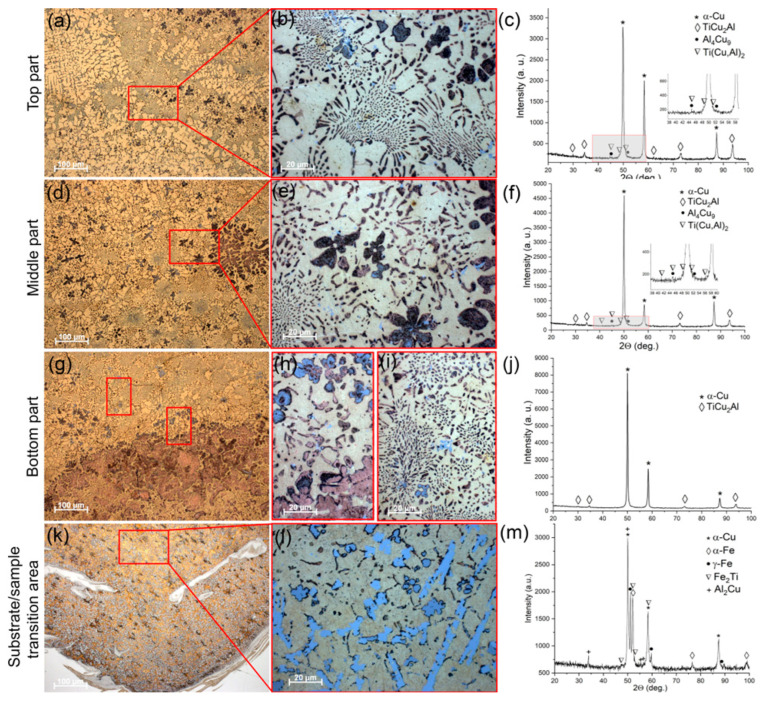
Optical images of macro- and microstructures found in the top (**a**,**b**), middle (**d**,**e**), and bottom (**g**,**h**,**i**) parts of the bronze/10%Ti wall and the transition zone (**k**,**l**) with corresponding XRD patterns (**c**,**f**,**i**,**j**,**m**).

**Figure 6 materials-16-04279-f006:**
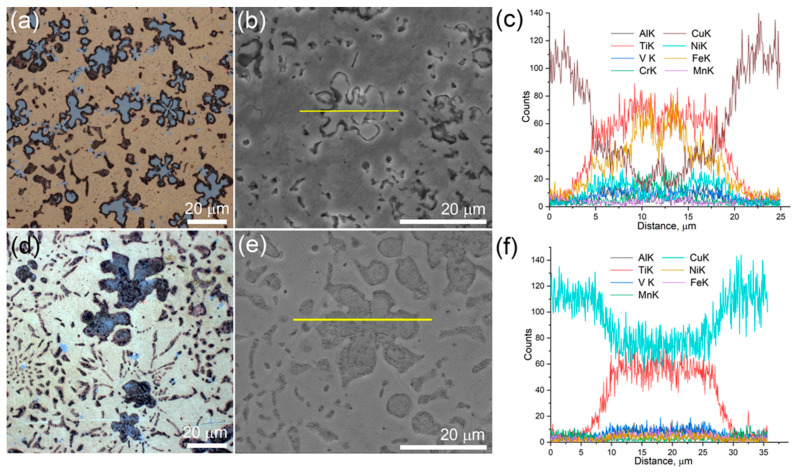
Flower-like dendrites in the transition zone of the bronze/10%Ti (**a**,**b**) and in the middle part of the wall (**d**,**e**) with corresponding EDS elemental profiles (**c**,**f**), respectively.

**Figure 7 materials-16-04279-f007:**
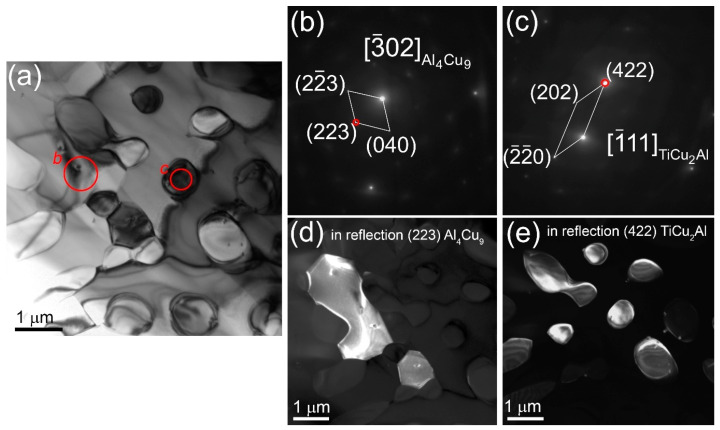
TEM bright-field image (**a**) SAED patterns (**b**,**c**) and dark-field images (**d**,**e**) obtained using reflections (223)_Al4Cu9_ and (422)_TiCu2Al_ indicated by red circles in the corresponding SAED patterns particles.

**Figure 8 materials-16-04279-f008:**
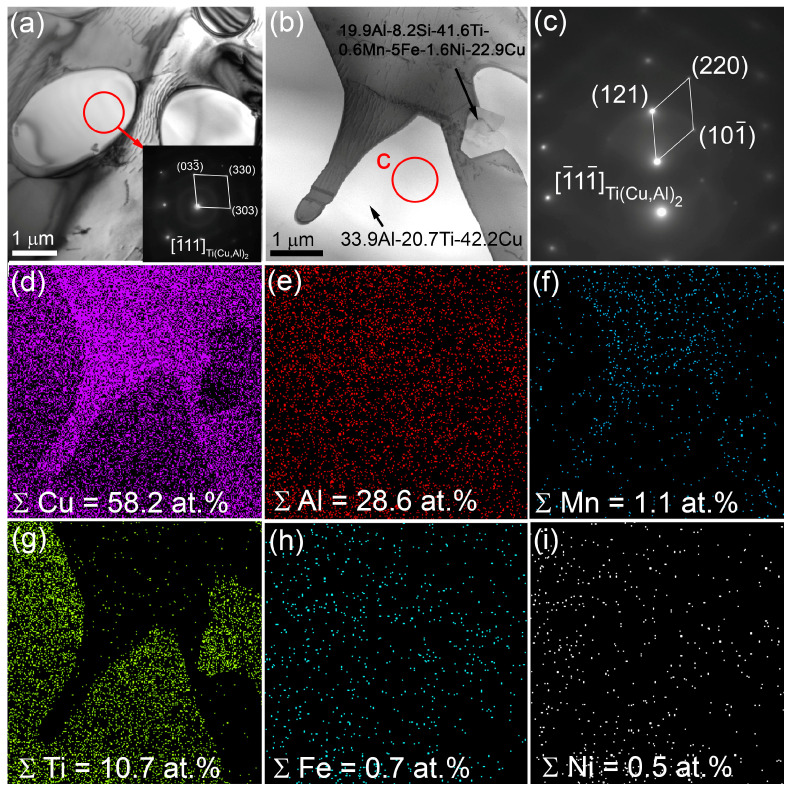
TEM bright-field (**a**,**b**) images of the bronze/10%Ti alloy, SAED pattern (**c**) and the corresponding EDS elemental maps (**d**–**i**).

**Figure 9 materials-16-04279-f009:**
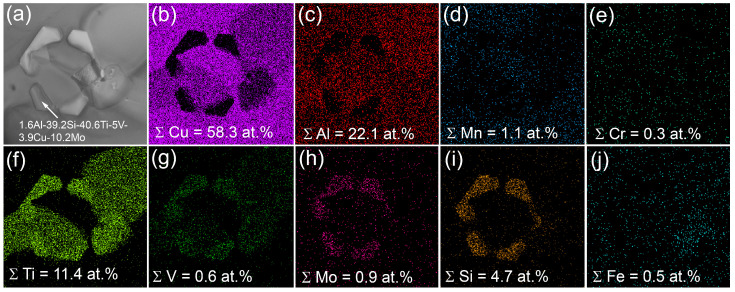
TEM bright-field (**a**) image of a partially dissolved particle in bronze/10%Ti alloy and corresponding EDS elemental maps (**b**–**j**).

**Figure 10 materials-16-04279-f010:**
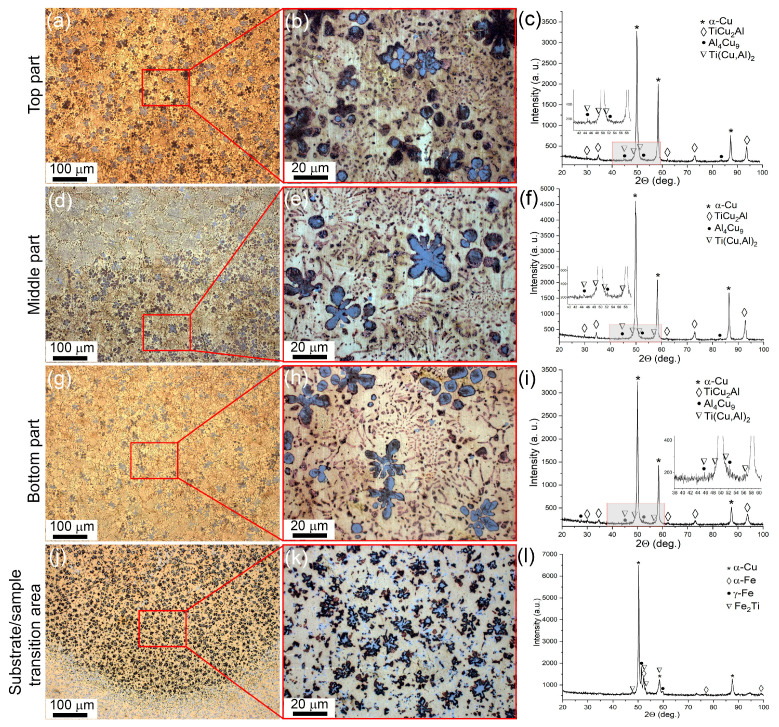
Optical images of macro- and microstructures found in the top (**a**,**b**), middle (**d**,**e**), and bottom (**g**,**h**) parts of the bronze/15%Ti wall and the transition zone (**j**,**k**) with corresponding XRD patterns (**c**,**f**,**i**,**l**).

**Figure 11 materials-16-04279-f011:**
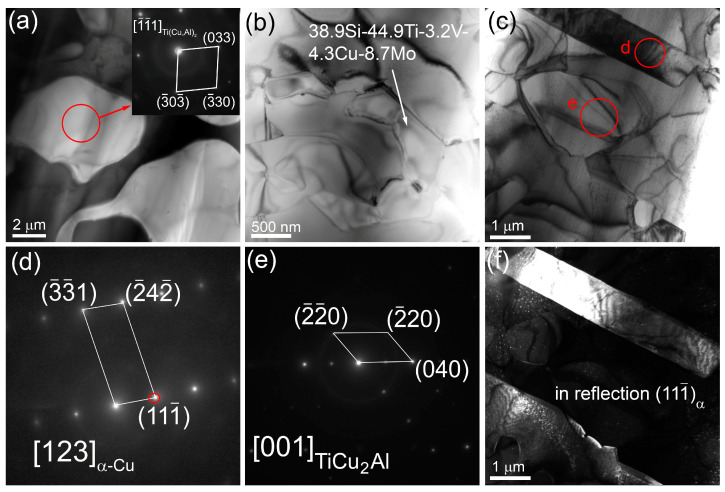
TEM bright-field (**a**–**c**) images of microstructures in the middle part of the bronze/10%Ti alloy wall, SAED patterns (**d**,**e**) and dark-field image (**f**) of annealing twin in the corresponding α-Cu grain) obtained using the ($11\bar{1}$)_α_ reflection indicated by the red circle in (**d**).

**Figure 12 materials-16-04279-f012:**
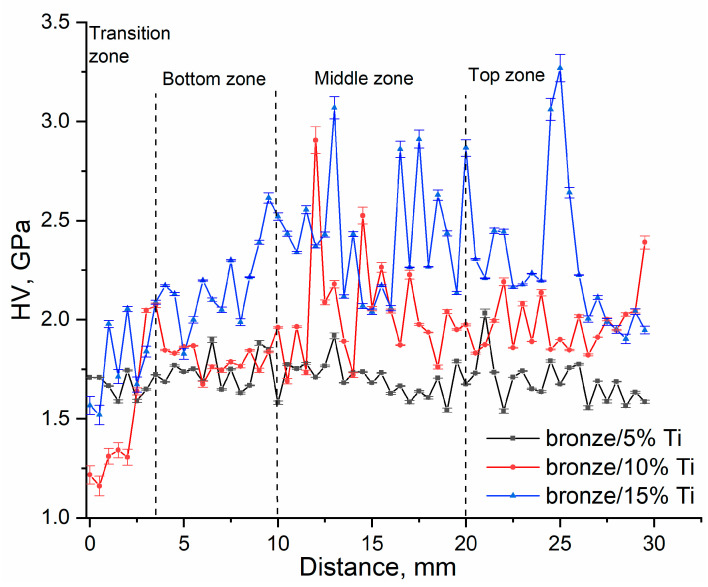
Microhardness profiles obtained along the Z axis (the wall growth direction).

**Figure 13 materials-16-04279-f013:**
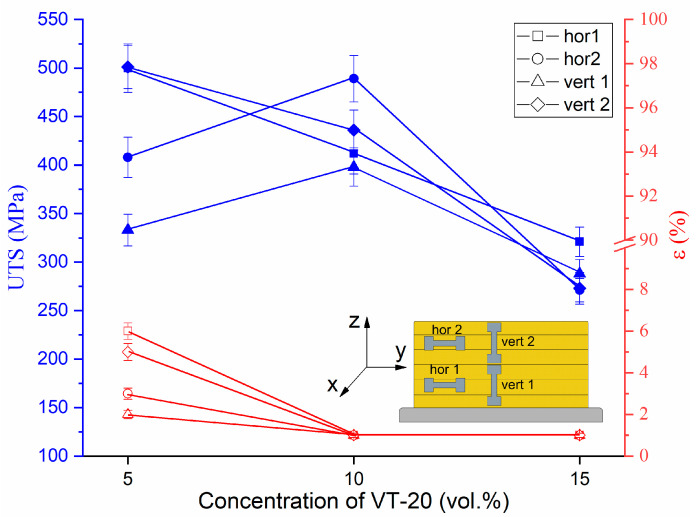
The dependencies of ultimate tensile strength and relative elongation on the titanium alloy percentage in the CuAl9Mn2/VT-20 alloy.

**Figure 14 materials-16-04279-f014:**
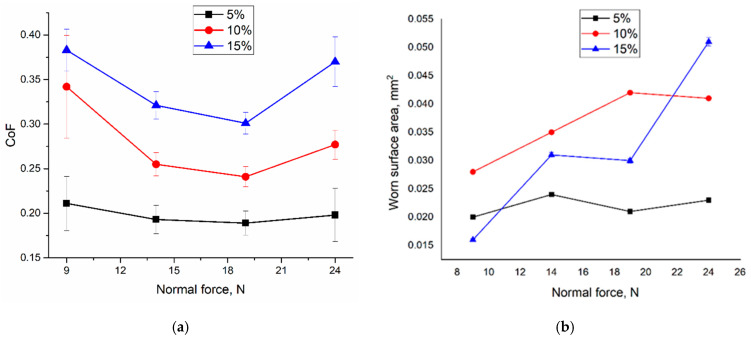
Mean CoF values (**a**) and vibration acceleration amplitudes (**b**) of CuAl9Mn2/VT-20 alloy samples rubbed by a Si_3_N_4_ ball as functions of normal load.

**Table 1 materials-16-04279-t001:** Chemical compositions of wires and CuAl9Mn2/VT-20 alloys.

Material	Sample Part	Elements, wt.%									
		Cu	Al	Mn	Fe	Zn	Ti	V	Mo	Zr	Ni
CuAl9Mn2	-	bal.	9.3 ± 1.4	1.9 ± 0.04	0.3 ± 0.01	0.4 ± 0.02	-	-	-	-	-
VT-20	-	0.04 ± 0.01	3.7 ± 0.5	-	0.07 ± 0.02	-	91.8 ± 0.6	1.2 ± 0.1	1.4 ± 0.02	1.7 ± 0.02	-
bronze/5%Ti	bottom	88.6 ± 0.6	6.5 ± 0.6	1.6 ± 0.03	0.3 ± 0.01	0.4 ± 0.02	2.2 ± 0.04	0.08 ± 0.02	0.04 ± 0.003	0.04 ± 0.003	0.3 ± 0.01
	middle	87.7 ± 0.6	7.4 ± 0.6	1.6 ± 0.03	0.3 ± 0.01	0.3 ± 0.02	2.2 ± 0.04	0.05 ± 0.01	0.04 ± 0.003	0.04 ± 0.003	0.3 ± 0.01
	top	87.2 ± 0.6	7.9 ± 0.6	1.6 ± 0.03	0.3 ± 0.01	0.4 ± 0.02	2.3 ± 0.04	0.07 ± 0.02	0.04 ± 0.003	0.04 ± 0.003	0.3 ± 0.01
bronze/10%Ti	bottom	85.6 ± 0.5	6.2 ± 0.5	1.6 ± 0.03	0.3 ± 0.01	0.4 ± 0.02	5.4 ± 0.07	0.1 ± 0.03	0.08 ± 0.004	0.09 ± 0.004	0.3 ± 0.02
	middle	86.0 ± 0.6	5.4 ± 0.6	1.6 ± 0.03	0.3 ± 0.01	0.4 ± 0.02	5.7 ± 0.08	0.1 ± 0.03	0.09 ± 0.004	0.1 ± 0.005	0.3 ± 0.02
	top	84.9 ± 0.5	6.5 ± 0.5	1.6 ± 0.03	0.3 ± 0.01	0.3 ± 0.02	5.7 ± 0.07	0.1 ± 0.02	0.08 ± 0.004	0.1 ± 0.005	0.3 ± 0.02
bronze/15%Ti	bottom	83.7 ± 0.6	6.1 ± 0.6	1.5 ± 0.03	0.4 ± 0.02	0.3 ± 0.02	7.3 ± 0.08	0.1 ± 0.03	0.1 ± 0.004	0.1 ± 0.005	0.2 ± 0.02
	middle	84.3 ± 0.5	6.4 ± 0.5	1.5 ± 0.03	0.3 ± 0.02	0.3 ± 0.02	6.5 ± 0.08	0.1 ± 0.03	0.09 ± 0.004	0.1 ± 0.005	0.2 ± 0.02
	top	83.9 ± 0.5	5.9 ± 0.5	1.5 ± 0.03	0.3 ± 0.02	0.3 ± 0.02	7.3 ± 0.07	0.1 ± 0.03	0.1 ± 0.004	0.1 ± 0.005	0.2 ± 0.02

**Table 2 materials-16-04279-t002:** EDS spectra of probe zones shown in Figure 3d. (Figure 3 and Figure 4).

Spectrum	Element, at.%					
	Al	Ti	Cu	Mn	Fe	Ni
1 (Figure 3d)	28.9	21.5	49.5	-	-	-
2 (Figure 3d)	27.3	36.6	30.1	-	5.9	-
3 (Figure 3d)	37.4	21.0	41.6	-	-	-
4 (Figure 3d)	31.4	19.2	49.4	-	-	-
5 (Figure 3d)	20.3	-	77.9	1.8	-	-
6 (Figure 3e)	39.4	19.7	40.9	-	-	-
7 (Figure 4a)	34.7	20.5	38.8	-	3.4	2.6
8 (Figure 4a)	32.8	-	67.2	-	-	-
9 (Figure 4a)	36.9	-	63.1	-	-	-

## Data Availability

The data presented in this study are available on request from the corresponding author.

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
