# Peer review of "Microstructures and Phases in Electron Beam Additively Manufactured Ti-Al-Mo-Zr-V/CuAl9Mn2 Alloy"

_materials, 2023, doi:10.3390/ma16124279_

Round 1
Reviewer 1 Report
In this manuscript, the authors studied the microstructures and phases in electron beam additively manufactured Ti-Al-Mo-Z-V/CuAl9Mn2 composite alloy. Overall, the paper contains good findings but there is a room for further improvement:
1- How many times did the authors repeated the tensile testing?
2- There are a great deal of published reports on additive manufacturing of metallic composite parts; the authors are required to go through the following example papers and cite them in order to provide link between the current work and previously published findings:
- Improvement in the high-temperature creep properties via heat treatment of Ti-6Al-4V alloy manufactured by selective laser melting
- High temperature tribology and wear of selective laser melted (SLM) 316L stainless steel
- Thermal behavior of the molten pool, microstructural evolution, and tribological performance during selective laser melting of TiC/316L stainless steel nanocomposites: Experimental and simulation methods
- Tribological properties of selective laser melted Al12Si alloy
- Tribological behavior of graphene-reinforced 316L stainless-steel composite prepared via selective laser melting
- Effect of Laser Mode and Power on the Tribological Behavior of Additively Manufactured IN718 Alloy
- Effect of process parameters on tribological performance of 316L stainless steel parts fabricated by selective laser melting
Author Response
Dear Ms./Mr. Reviewers
We appreciate very much all comments from the reviewers and did our best to provide answers to all of them. Also, we made revisions to the manuscript text in response to the comments and highlighted points.
On behalf of all co-authors,
Andrey Chumaevskii
In this manuscript, the authors studied the microstructures and phases in electron beam additively manufactured Ti-Al-Mo-Z-V/CuAl9Mn2 composite alloy. Overall, the paper contains good findings but there is a room for further improvement:
1- How many times did the authors repeated the tensile testing?
A: Three specimens were tested for each experimental datapoint.
2- There are a great deal of published reports on additive manufacturing of metallic composite parts; the authors are required to go through the following example papers and cite them in order to provide link between the current work and previously published findings:
- Improvement in the high-temperature creep properties via heat treatment of Ti-6Al-4V alloy manufactured by selective laser melting
- High temperature tribology and wear of selective laser melted (SLM) 316L stainless steel
- Thermal behavior of the molten pool, microstructural evolution, and tribological performance during selective laser melting of TiC/316L stainless steel nanocomposites: Experimental and simulation methods
- Tribological properties of selective laser melted Al12Si alloy
- Tribological behavior of graphene-reinforced 316L stainless-steel composite prepared via selective laser melting
- Effect of Laser Mode and Power on the Tribological Behavior of Additively Manufactured IN718 Alloy
- Effect of process parameters on tribological performance of 316L stainless steel parts fabricated by selective laser melting
A: There is large difference between selective laser melting on stainless steel and electron beam wire additive manufacturing of titanium-bronze alloys and we can not see how these references can provide a link between the current work and previously published findings.

Reviewer 2 Report
In this work, the effect of titanium additives on the evolution of the structural-phase composition, mechanical and tribological properties of Ti-Al-Mo-Z-V/CuAl9Mn2 alloy have been investigated.
Abstract :It is well written. The findings are concurrent and justified. The idea is novel and has important industrial application in additive manufacturing. What is selection reason as Ti with different composition rather Al alloy composition is higher.
Introduction : It is well written, and literature is justified.
Result and discussion . Authors have presented many useful findings, but they are not clearly explained? And how these results are compared with existing literature and justified?. I would suggest adding relevant referecnes in all subsections.
Conclusion : It looks too short as result and discussion is quite lengthy.
Author Response
Dear Ms./Mr. Reviewer
We appreciate very much all comments from the reviewers and did our best to provide answers to all of them. Also, we made revisions to the manuscript text in response to the comments and highlighted points.
On behalf of all co-authors,
Andrey Chumaevskii
In this work, the effect of titanium additives on the evolution of the structural-phase composition, mechanical and tribological properties of Ti-Al-Mo-Z-V/CuAl9Mn2 alloy have been investigated.
Abstract: It is well written. The findings are concurrent and justified. The idea is novel and has important industrial application in additive manufacturing. What is selection reason as Ti with different composition rather Al alloy composition is higher.
A: Ti alloys have higher resilience as compared to the Al-based ones.
Introduction: It is well written, and literature is justified.
A: Thank you.
Result and discussion . Authors have presented many useful findings, but they are not clearly explained? And how these results are compared with existing literature and justified?. I would suggest adding relevant referecnes in all subsections.
A: Thank you. References added.
Conclusion : It looks too short as result and discussion is quite lengthy.
A: Conclusion section was extended

Reviewer 3 Report
I want to compliment the authors for their interesting and rigorous research work.
Some comments from my side:
TITLE:
* "composite alloy": why are you using "composite"? A composite is characterized by different components which are not different phases: they are separated by a sharp interface that is not a grain boundary. This is not the case in my opinion.
Revise the use of the term "composite" in the whole paper.
ABSTRACT:
* "steady oxidation wear": this form is not universally known: explain better the behavior of the material on sliding. The same must be done in the discussion.
INTRODUCTION
You have some sentences referring to bronze and Cu-based alloys, then you move to Ti-based alloys and Al-based alloys: make it much more clear which is the majority element of each alloy you are speaking about.
Pay attention to the use of "load" (that refers to MPa) and "force" (that refers to N).
A mention of additive manufacturing of Titanium alloys could be important: look at https://doi.org/10.3390/met11091343 or any other paper dealing with this topic.
MATERIALS AND METHODS:
* Tribological test: Were the tests performed with any lubricant? add something about this.
Results
* Some typos: page 4 line 145-146 "и"
* "The volume percentages of (α + Al4Cu9 + TiCu2Al) eutectics in top, middle and bottom 163 parts of the wall are 15, 20 and 25%, respectively": it is not clear which are the other phases to reach 100%.
* Figure 13: It is not clear if there is any difference between Hor 1 and Hor 2 or Vert 1 and Vert2
* I suggest mentioning that the counterpart was Si3N4, even if it was mentioned in the materials and methods section, and explain why did you select this counter material.
Author Response
Dear Ms./Mr. Reviewer
We appreciate very much all comments from the reviewers and did our best to provide answers to all of them. Also, we made revisions to the manuscript text in response to the comments and highlighted points.
On behalf of all co-authors,
Andrey Chumaevskii
I want to compliment the authors for their interesting and rigorous research work.
Some comments from my side:
TITLE:
* "composite alloy": why are you using "composite"? A composite is characterized by different components which are not different phases: they are separated by a sharp interface that is not a grain boundary. This is not the case in my opinion.
Revise the use of the term "composite" in the whole paper.
A: Thank you. Albeit being somewhat awkward this term was supposed to denote a dual nature of such an alloy, i.e. it is an alloy (which may be a composite by itself) additionally reinforced by in-situ formed intermetallic particles from alloying by dissimilar metal. Despite that, we changed the term throughout the text to read “alloy”.
ABSTRACT:
* "steady oxidation wear": this form is not universally known: explain better the behavior of the material on sliding. The same must be done in the discussion.
A: Steady wear is a wear regime that is characterized by the absence of sharp wear peaks and troughs, i.e. when worn surface is covered by oxide films, which usually reduce both wear and friction as well as stabilize the wear rate.
INTRODUCTION
You have some sentences referring to bronze and Cu-based alloys, then you move to Ti-based alloys and Al-based alloys: make it much more clear which is the majority element of each alloy you are speaking about.
A: Corrected
Pay attention to the use of "load" (that refers to MPa) and "force" (that refers to N).
A mention of additive manufacturing of Titanium alloys could be important: look at https://doi.org/10.3390/met11091343 or any other paper dealing with this topic.
A: Corrected
MATERIALS AND METHODS:
* Tribological test: Were the tests performed with any lubricant? add something about this.
A: Thank you. The following sentence was added to the text: “Tribological testing was carried out using a“ball-on-disk scheme” when a Si3N4 ∅6 mm ball was sliding on a disk machined from the composite alloy wall without adding any lubricant”.
Results
* Some typos: page 4 line 145-146 "и"
A: Thank you. Corrected.
* "The volume percentages of (α + Al4Cu9 + TiCu2Al) eutectics in top, middle and bottom 163 parts of the wall are 15, 20 and 25%, respectively": it is not clear which are the other phases to reach 100%.
A: Thank you. The Bronze/5%Ti alloy is composed of aluminum bronze matrix and (α + Al4Cu9 + TiCu2Al) eutectics.
* Figure 13: It is not clear if there is any difference between Hor 1 and Hor 2 or Vert 1 and Vert2
A: Thank you. The tensile test strength anisotropy description has been added to corresponding section 3.4.
* I suggest mentioning that the counterpart was Si3N4, even if it was mentioned in the materials and methods section, and explain why did you select this counter material.
A: Thank you. This hard and chemically inert material was selected because of being composed of elements that did not present in both alloys in order to easily detect its wear particles and differentiate them from others.

Reviewer 4 Report
Authors are asked to emphasize more in detail the novelty of their work. Also, the number of references is rather small. Authors should discuss more in detail the current status of the topic. Moreover, in the case of references, I would also recommend higher diversity.
Scale bars in Figure 2 and Figure 5 should be enlarged.
The elemental mapping in Figure 8 and Figure 9 are not the best quality. Are the results in Figure 8efhi reliable? E.g. it is difficult to see the difference between Al and Fe, which according to Authors is 28.6% to 0.7%.
Author Response
Dear Ms./Mr. Reviewer
We appreciate very much all comments from the reviewers and did our best to provide answers to all of them. Also, we made revisions to the manuscript text in response to the comments and highlighted points.
On behalf of all co-authors,
Andrey Chumaevskii
Authors are asked to emphasize more in detail the novelty of their work. Also, the number of references is rather small. Authors should discuss more in detail the current status of the topic. Moreover, in the case of references, I would also recommend higher diversity.
A: There is not so much literature sources that directly refer to the additive manufacturing of the aluminum bronze – Ti alloy system. To the best of our knowledge, all of them were included. More references were added with Introduction section.
Scale bars in Figure 2 and Figure 5 should be enlarged.
A: Thank you. Corrected.
The elemental mapping in Figure 8 and Figure 9 are not the best quality. Are the results in Figure 8efhi reliable? E.g. it is difficult to see the difference between Al and Fe, which according to Authors is 28.6% to 0.7%.
A: Unfortunately, these images were obtained directly from the TEM EDS add-on and can not be further improved. The reliability of the results is determined by the EDS method accuracy which is rather high for detecting metals.
